# Genetic Deletion of Thorase Causes Purkinje Cell Loss and Impaired Motor Coordination Behavior

**DOI:** 10.3390/cells12162032

**Published:** 2023-08-10

**Authors:** Chao Li, Han Zhang, Kexin Tong, Menghua Cai, Fei Gao, Jia Yang, Yi Xu, Huaishan Wang, Hui Chen, Yu Hu, Wei He, Jianmin Zhang

**Affiliations:** 1Department of Immunology, CAMS Key Laboratory T Cell and Cancer Immunotherapy, Institute of Basic Medical Sciences, Chinese Academy of Medical Sciences and School of Basic Medicine, Peking Union Medical College, State Key Laboratory of Medical Molecular Biology, Beijing 100005, China; lichao1119@163.com (C.L.); hbzhanghan@ibms.pumc.edu.cn (H.Z.); tkx199746@163.com (K.T.); menghuacai@ibms.pumc.edu.cn (M.C.); gaofeipumc2015@163.com (F.G.); jia.yang@ucsf.edu (J.Y.); xuyi@ibms.pumc.edu.cn (Y.X.); huaishan.wang@pennmedicine.upenn.edu (H.W.); chenhui@ibms.pumc.edu.cn (H.C.); yu.hu@ibms.pumc.edu.cn (Y.H.); 2Changzhou Xitaihu Institute for Frontier Technology of Cell Therapy, Changzhou 213000, China; 3Haihe Laboratory of Cell Ecosystem, Chinese Academy of Medical Sciences and Peking Union Medical College, Tianjin 300010, China

**Keywords:** Thorase, cerebellum, Purkinje cells, neuroinflammation, neurodegenerative disease

## Abstract

Thorase belongs to the AAA+ ATPase family, which plays a critical role in maintaining cellular homeostasis. Our previous work reported that Thorase was highly expressed in brain tissue, especially in the cerebellum. However, the roles of Thorase in the cerebellum have still not been characterized. In this study, we generated conditional knockout mice (cKO) with Thorase deletion in Purkinje cells. Thorase cKO mice exhibited cerebellar degenerative diseases-like behavior and significant impairment in motor coordination. Thorase deletion resulted in more Purkinje neuron apoptosis, leading to Purkinje cell loss in the cerebellum of Thorase cKO mice. We also found enhanced expression of the inflammatory protein ASC, IL-1β, IL-6 and TNF-α in the Thorase cKO cerebellum, which contributed to the pathogenesis of cerebellar degenerative disease. Our findings provide a better understanding of the role of Thorase in the cerebellum, which is a theoretical basis for Thorase as a therapeutic drug target for neurodegenerative diseases.

## 1. Introduction

The cerebellum is one of the most important regions in the human brain and the largest motor structure in the central nervous system, which is responsible for maintaining body balance, regulating muscle tone, coordinating voluntary movements and participating in motor learning [1,2]. Cerebellum nonmotor functions include executive function, visuospatial processing, language function and emotion regulation [3,4,5]. The cerebellar ataxias comprise a heterogeneous group of inherited and acquired disorders that manifest as clumsiness or imbalance of movement due to decreased function of the cerebellum, often resulting from cerebellar Purkinje neuronal degeneration or a variety of genetic mutations and traumatic injury [6,7,8]. The various models of presentation of cerebellar ataxia includes at least 37 dominantly inherited spinocerebellar ataxias (SCAs), 20 recessive inherited ataxias and mitochondrial inherited forms of cerebellar ataxia [9,10,11]. For example, the dysregulation of mitochondrial electron transport chain activity was observed in SCA1 mouse cerebella [12] and mutations in AFG3L2 encoding AFG3-like AAA ATPase 2 and might result in mitochondrial function abnormal and subsequently cerebellar degeneration [13,14].

Purkinje neurons, a class of γ-aminobutyric acidergic neurons, are located in the cerebellar cortex and integrate all the input into the cerebellum [15,16,17]. These cells are one of the largest neurons in the human brain and are characterized by a large number of dendritic fibers that penetrate deep into the molecular layer. Purkinje cells are the main neurons and the only efferent neurons in the cerebellar cortex neuronal circuit, which constitutes the neuronal circuit responsible for sensory–motor integration in the cerebellar cortex [5,16,18]. A large number of studies have reported abnormal Purkinje cell morphology and function in cerebellar diseases, including cerebellar ataxia [6,19,20]. In mouse models, Purkinje cell deletions resulted in impairment in motor learning ability, movement disorders and ataxia [20]. In addition, neuroinflammation mediated by microglia and astrocyte activation also plays an important role in cerebellar neurodegenerative diseases [21,22,23].

Thorase is a member of the AAA+ ATPase family that is localized in the outer membrane of the mitochondria and plays an important role in maintaining healthy mitochondrial networks by participating in the regulation of mitochondrial quality control [24,25,26]. Our previous studies demonstrated that Thorase is involved in the formation of learning and memory by regulating the plasticity of the postsynaptic membrane [27,28]. Thorase variants were found in schizophrenia patients via bidirectional sequencing of exons of the gene encoding Thorase (ATAD1) [29]; another ATAD1 mutation was identified as the cause of severe encephalopathy and congenital stiffness [30,31,32]. In addition, the loss of Thorase caused a series of severe nervous system dysfunctions and aggravated the damage to the brain function of mice in a stroke model [28] and Parkinson’s disease [33].

Given the vital functions that Thorase plays in the nervous system and its high expression in the cerebellum, in this study, we characterized the phenotypes of mice with Thorase deletion in Purkinje cells. The results revealed that mice lacking Thorase in Purkinje cells exhibited cerebellar neurodegenerative behaviors and significant inflammatory pathogenesis in the cerebellum.

## 2. Materials and Methods

### 2.1. Experimental Animals

All animal procedures were approved by the Institutional Animal Care and Use Committee (IACUC) of CAMS. Thorase^+/−^ mice and Thorase^flox/+^ mice were gifted by Drs. Dawson and Valina Dawson from the Institute of Cell Engineering, John Hopkins University School of Medicine. The original Pcp2-iCre^+^ transgenic mice, which were specifically expressed by Purkinje cerebellum cells, were purchased from Nanjing Model Animal Center and bred at the Institute of Experimental Animals of CAMS.

The brain tissues used in Western blotting were collected from 8-month-old Thorase cKO mice (*n* = 5) and WT littermates (*n* = 5) and 12-month-old Thorase cKO mice (*n* = 4) and WT littermates (*n* = 4). For ELISA, we used the brain tissue from 12-month-old Thorase cKO mice (*n* = 4) and WT littermates (*n* = 4). For immune staining, the brain tissue sections were from 3-month-old Thorase cKO mice (*n* = 3) and WT littermates (*n* = 3), 8-month-old Thorase cKO mice (*n* = 5) and WT littermates (*n* = 5), and 12-month-old Thorase cKO mice (*n* = 4) and WT littermates (*n* = 4). For electron microscopy observation, we used the brain tissue isolated from 8-month-old Thorase cKO mice (*n* = 5) and WT littermates (*n* = 5).

### 2.2. Antibodies and Reagents

The following antibodies were used in the study: mouse monoclonal anti-calbindin antibody (Millipore, Darmstadt, Germany, 1:1000), rabbit monoclonal anti-ASC antibody (Adipogen, Beijing, China, 1:1000), anti-Thorase (Biolegend, San Diego, CA, USA, 1:1000) and anti-Iba1 (Wako, Tokyo, Japan, 1:1000), rabbit anti-GFAP (Abcam, Cambridge, UK, 1:1000) and mouse monoclonal β-actin (BOSTER, Wuhan, China, 1:1000). Horseradish peroxidase (HRP)-labeled goat anti-mouse and goat anti-rabbit antibodies (ZSGB, Beijing, China, 1:5000), Alexa Fluor^®^ 488-labeled goat anti-mouse and Alexa Fluor^®^ 555-labeled goat anti-rabbit antibodies (CST, Danvers, MA, USA, 1:2000) were also used.

The following reagents were used in the study: SuperSignal West Pico PLUS Chemiluminescent Substrate (Thermo Fisher Scientific, Waltham, MA, USA), DAB kit (ZSGB, Beijing, China) and O.C.T. Compound (Tissue-Tek, Sakura, CA, USA).

### 2.3. Mouse Genotyping

The genotypes of mice were determined using PCR according to the procedure as described previously [34]. The primers for mouse genotyping were as follows:

Primers for genotyping Thorase^+/−^ and Thorase^flox/+^: JZ129 5′- GCA TGT GCA ACT GAA CGC AGG G -3′, JZ156 5′- CCA CCA CCC ACT TTG TTT TGA AGT TAA -3′, JZ178 5′- GGA GCT TTC AGT CAG TAC CAT TTC ACT TTC -3′.

Primers for genotyping Pcp2-iCre were as follows: Pcp2F 5′- GCT TGC ATT ACC GGT CGA TGC -3′, Pcp2R 5′- CAG GGT GTT ATA AGC AAT CCC -3′; OMIR7338 5′- CTA GGC CAC AGA ATT GAA AGA TCT -3′, OMIR7339 5′- GTA GGT GGA AAT TCT AGC ATC ATC C-3′.

### 2.4. Behavioral Measurements

#### 2.4.1. Rotarod Test

Experiments were performed using 8-month-old pcp2-Cre+ Thorase cKO mice and their littermate wild-type mice (*n* = 7 males and 7 females) in five days as a standard experimental procedure [35,36]. The first three days were mouse learning time, and days 4–5 were used for experimental tests. Each mouse had a total duration of 300 s per experiment with a starting speed of 5 rpm, and the speed of the rod was accelerated evenly to 40 rpm within 180 s and maintained at 120 s. The experiment was repeated three times a day for 5 min per animal, and the next measurements were performed at an interval of 30 min. The longest latency to falling of each mouse each day was recorded and analyzed. 

#### 2.4.2. Footprint Test

The experiment was performed using a homemade device that measured stride length and step spacing of the front and back limbs of mice according to the standard experimental procedure [37]. Briefly, 12-month-old cKO mice and their littermate WT mice (*n* = 3 males and 3 females) were subjected to the experiments. The front and back paws of mice are coated with nontoxic ink in red and black, respectively. A 60 W light bulb was placed at the beginning of the dark tunnel (10 × 10 × 50 cm), and a dark cage was placed at the end. Mice were trained twice a day for three days before the experiment, and the mice were required to walk along a narrow dark tunnel from the starting point. During the formal experiment, a black piece of measuring paper is laid under the tunnel to record the footprints.

### 2.5. Tissue Preparation

The cKO mice and littermate WT mice at 3, 8 and 12 months of age were anesthetized, and perfused with PBS solution for Western blotting, or with 4% PFA for paraffin sections. For Western blotting, the cerebellum tissue was placed in liquid nitrogen for quick freezing and stored at −80 °C. The brain tissue was fixed with 4% paraformaldehyde for 48 h and dehydrated in 30% sucrose solution until the brain tissue sank to the bottom. Frozen brain tissue sections were embedded in O.C.T. Compound (Tissue-Tek, Sakura, Japan) and cut into 25–30 μm using Leica CM1950 cryostats (Bensheim, Germany). For paraffin slices, the tissues were paraffin-embedded, sectioned into 5 μm sections and stored at room temperature until used for immunohistochemistry, immunofluorescence staining and electron microscopy observation. 

### 2.6. Western Blot Analysis

Brain tissue was extracted with RIPA lysis buffer, and protein concentrations were determined using the BCA method. We utilized approximately 20 microg of protein for SDS-PAGE separation and then transferred the protein to a membrane. After blocking with 5% skimmed milk powder for 2 h, the primary antibody was added overnight at 4 °C. Then, after washing with TBST three times and adding HRP-labeled goat anti-rabbit or mouse secondary antibody at room temperature for 1 h, we used SuperSignal West Pico PLUS Chemiluminescent Substrate (Thermo Fisher Scientific) to detect horseradish peroxidase (HRP) on immunoblots. The band grayscale value was analyzed, and the relative expression level of the targeted protein was statistically analyzed using the ratio of the targeted protein to the endogenous reference protein.

### 2.7. Immunohistochemistry (IHC) Staining

Paraffin sections were baked, after high-temperature antigens repaired and endogenous peroxidase blocked, membranes were broken with 0.25% Triton X-100 at room temperature for 20 min. A circle was drawn outside the tissue with an immunohistochemical oil pen, and the ready-to-use 5% goat serum was added dropwise in the circle and closed at room temperature in a wet box for 2 h. After adding primary antibody to the slice tissue, the slices were incubated overnight at 4 °C. TBST buffer was used to wash five times, and the ImmPRESS^TM^ Polymer Reagent Peroxidase secondary antibody solution was added to the slices at room temperature for 1 h. After DAB staining and hematoxylin counterstaining, the slices were dehydrated before mounting. The Leica DM6 B microscope was used to scan the slices.

### 2.8. Immunofluorescence (IF) Staining

The brain tissue slices were warmed to room temperature and dried at 37 °C for 60 min. Then they were permeabilized with 0.25% Triton X-100 for 20 min at room temperature. We added the ready-to-use 5% goat serum dropwise to the slices and placed them at room temperature in a wet box for 2 h. After adding primary antibody to the slice tissue, the slices were incubated overnight at 4 °C. TBST buffer was used to wash the tissues five times, and the Alexa Fluor^®^ 488-labeled or 555-labeled anti-mouse or anti-rabbit secondary antibody solution was added to the slices at room temperature and protected from light for 1 h. Then, the slices were incubated with DAPI nucleic acid dye at room temperature and protected from light for 10 min. After TBST washing three times, the slices were mounted and observed using Zeiss LSM780 confocal fluorescence microscopy.

### 2.9. Enzyme-Linked Immunosorbent Assay (ELISA)

The concentrations of IL-1β, IL-6 and TNF-α in the brain tissue samples of 12-month-old mice (*n* = 4) were measured according to the ELISA kit instructions and the standard experimental procedure [38].

### 2.10. Transmission Electron Microscopy

The mice were anesthetized and perfused with prechilled 0.2 M PB buffer, until they were completely fixed. We dissected the whole brain tissues of mice and placed them in the dish containing 2.5% glutaraldehyde. Then, we took the cerebellar tissues and cut them into pieces of 1 mm × 1 mm × 2 mm. The tissues were placed in 2.5% glutaraldehyde overnight at 4 °C. 0.2 M PB buffer was used to wash three times; then, the tissues were stained with 1% osmium acid for 1.5 h until the samples turned black. We washed the tissues three times with 0.2 M PB buffer and placed them in acetone at 50%, 70%, 80% and 90% for 5 min each, and in pure acetone twice for 10 min each. After embedding the tissues, we used an ultrathin slicer to continuously cut out 60–70 nm sections. Then, the sections were stained with lead citrate for 10 min and uranyl acetate for 30 min. Then, we used the transmission electron microscopy JEOL JEM-2100 to observe. 

### 2.11. Statistical Analysis

The data obtained in the experiment were plotted and statistically analyzed using Microsoft Office Excel Software 16.74 version. The result was expressed as the mean ± S.E.M. and the intergroup comparison was performed using the unpaired Student’s *t* test. Multigroup comparisons were performed using one-way ANOVA. Significant differences are presented as n.s., (not significant), *, *p* < 0.05, and **, *p* < 0.01. 

## 3. Results

### 3.1. Thorase cKO Mice Exhibited Impaired Motor Coordination

Our previous study demonstrated that there was a very high level of Thorase expression in Purkinje cells [27], which are GABA neurons located between the cerebellum granular layer and the molecular layer. To study the role of Thorase in the cerebellum, we prepared Thorase conditional knockout (cKO) mice in Purkinje cells by crossing Thorase^floxed/+^ mice with Pcp2-Cre^+^ mice. After two rounds of mating, we obtained Thorase^floxed/floxed^ × Pcp2-Cre^+^ mice, also named Thorase cKO mice. The homozygous genotype Thorase^floxed/floxed^ × Pcp2-Cre^−^ mice served as wild-type (WT) controls (Appendix A). Thorase cKO mice were verified using immunohistochemical staining with a Thorase antibody. No Thorase expression was observed in Purkinje cells (Appendix A).

Then, we applied the rotarod test to assess the motor coordination of Thorase cKO mice at the age of 8 months. Although in the first three days no significant difference was observed between WT and cKO mice, the movement time on the rod of Thorase cKO mice was significantly lower than those of littermate WT mice on the day 4 and day 5 (Figure 1A). Meanwhile, Thorase cKO mice showed a phenomenon of holding rods in the limbs earlier during the experiment. This phenomenon indicates that cKO mice do not learn to or have a strength defect to stay on the rod. We then examined the footprint of mice at the age of 12 months. The results showed that the movement stride length of the forelimbs and hindlimbs of Thorase cKO mice was significantly shorter than that of the WT control mice (Figure 1B–D). In addition, the step width was significantly narrower (Figure 1B). Together, these behavioral phenotypes suggest that the motor coordination ability of Thorase cKO mice was impaired, exhibiting characteristics of cerebellar motor neurodegenerative diseases.

### 3.2. Thorase Deficiency Causes Loss and Degeneration of Cerebellar Purkinje Cells

To investigate the effect of Thorase on cerebellar Purkinje cells, we examined morphological changes in the cerebellum of Thorase cKO mice using immunohistochemical staining and immunofluorescence staining. The results showed that although the number of Purkinje neurons in 3-month-old Thorase cKO mice was not significantly different from that in WT mice, the distal dendritic fibers were obviously missing (Figure 2A,B). The number of Purkinje neurons in 8- and 12-month-old Thorase cKO mice was significantly lower than that in WT mice, and the structure of dendritic fibers was seriously damaged (Figure 2A,B). These findings indicate that Thorase deficiency leads to the loss of cerebellar Purkinje cells. The dendritic spine structure of the Purkinje cell layer had almost disappeared and a large number of vacuole structures had appeared, which was seriously damaged (Figure 2B). We also performed transmission electron microscopy (TEM) to examine the subcellular changes in Purkinje cells. The results showed that the morphologies of organelles in Purkinje cells from cKO mice were altered, especially the mitochondria which showed more rounded and a higher mitochondrial density than those in WT Purkinje cells (Figure 2C). The mitochondria in cKO Purkinje cells were enlarged in size with a lower aspect ratio, and the mitochondrial matrix was swollen and had more vacuoles (Figure 2C). Taken together, these results suggest that Thorase deletion results in the loss and degeneration of Purkinje cells in the cerebellum of Thorase cKO mice.

### 3.3. Thorase Deficiency Causes Activation of Inflammatory Cells in Cerebellum

Previous studies have demonstrated that Purkinje cell apoptosis causes secondary inflammatory response in the Purkinje cell conditional knockouts in AFG3L2 mouse models [39]. Thus, we measured the distribution of activated microglia and astrocytes in the cerebellum regions from cKO mice and WT littermates using immunofluorescence staining with antibodies against Iba1 and glia fibroic acid protein (GFAP). The results showed that 3-month-old Thorase cKO mice exhibited slightly higher numbers of active microglia and astrocytes in the cerebellar molecular layer region than the of WT control mice (Figure 3A). However, in 8-month-old cKO mice, activated microglia and astrocyte were obviously higher than those in WT mice (Figure 3B). This finding was also validated using Western blotting. The expression of Iba1 and GFAP in brain tissues of 8-month-old Thorase cKO mice was significantly higher than that of their WT littermates (Figure 3C–E). In 12-month-old mice, we also observed the same phenomenon and a slight elevation of apoptosis-associated speck-like protein containing a CARD (ASC) protein expression (Figure 3F–I). Considering the small proportion of microglia, the difference in ASC protein may be masked by the background of other cell types, and immunofluorescence staining was used for further verification. The results showed that ASC spots appeared around the nucleus of cerebellar microglia of 8-month-old Thorase cKO mice, but not in WT littermates (Figure 3J). These results suggest that Thorase deficiency causes a cerebellar inflammatory response.

### 3.4. Thorase Deficiency Causes an Increase in Inflammatory Cytokines in Cerebellum

Then, we used enzyme-linked immunosorbent assays (ELISA) to detect the levels of the inflammatory cytokines TNF-α, IL-6 and IL-1β in cerebellar lysates from 12-month-old mice. Compared to WT littermates, TNF-α, IL-6 and IL-1β levels were significantly elevated in the cerebellum of 12-month-old Thorase cKO mice (Figure 4A–C). These results suggest that Thorase deficiency promotes neuroinflammation in the cerebellum of Thorase cKO mice.

## 4. Discussion

Purkinje neurons are highly sensitive to hypoxia, ischemia, traumatic injury and extrinsic toxicity, and excessive genetic mutations may directly cause Purkinje neuron degeneration and result in ataxia [40]. As the only output neurons of the cerebellar cortex that are vital in controlling motor functions, Purkinje neuron dysfunction may lead to a variety of cerebellar diseases, including dystonia, tremor, schizophrenia and autism spectrum disorder [41]. Purkinje cells have unique structure with large cell body and extensive dendritic branching, which require a high-level energy supplement and an accurate mitochondrial quality control [42]. Dysfunction of mitochondrial dynamics led to Purkinje cell degeneration in spinocerebellar ataxia type 3 (SCA3) mouse model [43].

Thorase belongs to the AAA ATPase family, which functions in a variety of cellular metabolism processes and may participate in regulating cell growth, survival and proliferation. Recently, Thorase was demonstrated to directly interact with mTOR at the interface between lysosomes and mitochondria, thereby disassembling and inactivating mTORC1 [44]. In addition, Thorase can decompose the GluA2/GRIP1 complex in the presence of ATP and is then involved in AMPAR-mediated nerve signaling [45]. Our previous studies confirmed that Thorase plays a vital role in removing misplaced α-synuclein from the outer membrane of mitochondria and plays a protective role in neurodegeneration mediated by α-synuclein [33]. Recently, several research groups have found that circRNA circ-ATAD1 (Thorase) plays a vital role in both solid tumors (such as gastric cancer, colorectal cancer and endometrial cancer) [46,47,48] and blood tumors (such as acute myeloid leukemia) [49]. Multi-omics results showed that downregulation of ATAD1 was a clinical biomarker for the pathological diagnosis and prognosis of prostate adenocarcinoma patients [50].

Previous studies have demonstrated that except for the high expression of Thorase in the hippocampus, substantia nigra and other parts, the expression of Thorase in Purkinje cells in the cerebellum was also very high [27,28]. Furthermore, Thorase is located in the outer membrane of the mitochondria and is demonstrated to regulate mitochondrial quality control, thereby maintaining healthy mitochondrial function [24,25,26]. Purkinje cells are the only efferent neurons in the cerebellar cortex neuronal circuit and play a vital role in motor balance and regulating muscle tone and mood [16,19]. In this study, we focused on characterizing the role of Thorase in Purkinje cells and the phenotypes of mice specifically lacking Thorase in Purkinje cells. 

The major finding of this study is the identification of Thorase as a key player in the survival of Purkinje cells and motor behavior as well as neuroinflammation in the cerebellum. Thorase cKO mice showed significantly impaired motor coordination, indicating that Purkinje cells might be severely damaged. Transmission electron microscopy analysis showed a damaged structure of dendritic fibers, a large number of vacuoles or apoptotic Purkinje cells in Thorase cKO mice. These findings were also verified using immunofluorescence staining showing the absence of dendritic fibers at the ends in the molecular layer. In addition, we also found that more microglia and astrocytes were activated in the cerebellum of Thorase cKO mice, indicating that Thorase deficiency results in activation of microglia and astrocytes and subsequently causes enhanced inflammation in the brain.

Previous studies have demonstrated that several potential mechanisms resulting in Purkinje cell loss included toxic protein misfolding and aggregation, transcriptional dysregulation and alterations in the calcium signaling. These may result in early synaptic neuronal deficits and progressive cerebellar dysfunction and further cause Purkinje cell loss and degeneration [42]. In this study, although we have not clarified the mechanism of the phenotypes of mice lacking Thorase in Purkinje cells, we speculated the possible mechanisms as follows: First, Thorase directly interacts with mTOR and is involved in the disassembly and inactivation of mTORC1, further affecting the relationship of mTOR–Raptor complex [44]. Thus, we will further explore whether Thorase deficiency causes the dysfunction of mTOR-GSK3β pathway and subsequently impairs mouse motor behavior in the cerebellum. Second, the possible types of programmed cell death including necroptosis may contribute to the loss of cerebellar Purkinje cells lacking Thorase. Third, the activation of microglia and astrocytes induced by damage-associated molecular patterns (DAMPs) results in an increased release of the inflammatory cytokines TNF-α, IL-1β and IL-6 in cerebellum, which mediates neurodegenerative diseases [51]. Further research in the molecular events would be beneficial to clarify the therapeutic targets for intervening in the progression of the cerebellar diseases.

## 5. Conclusions

In summary, the major aim of this study is to characterize the phenotypes of mice with genetic deletion of Thorase in Purkinje cells. We identified Thorase as a key player in the survival of Purkinje cells and the motor behavior as well as neuroinflammation in the cerebellum. However, we have not explored the mechanisms underlying the regulation of Thorase in the survival and synaptic plasticity of Purkinje cells as well as the behavior, which needs to be further investigated in our future studies.

## Figures and Tables

**Figure 1 cells-12-02032-f001:**
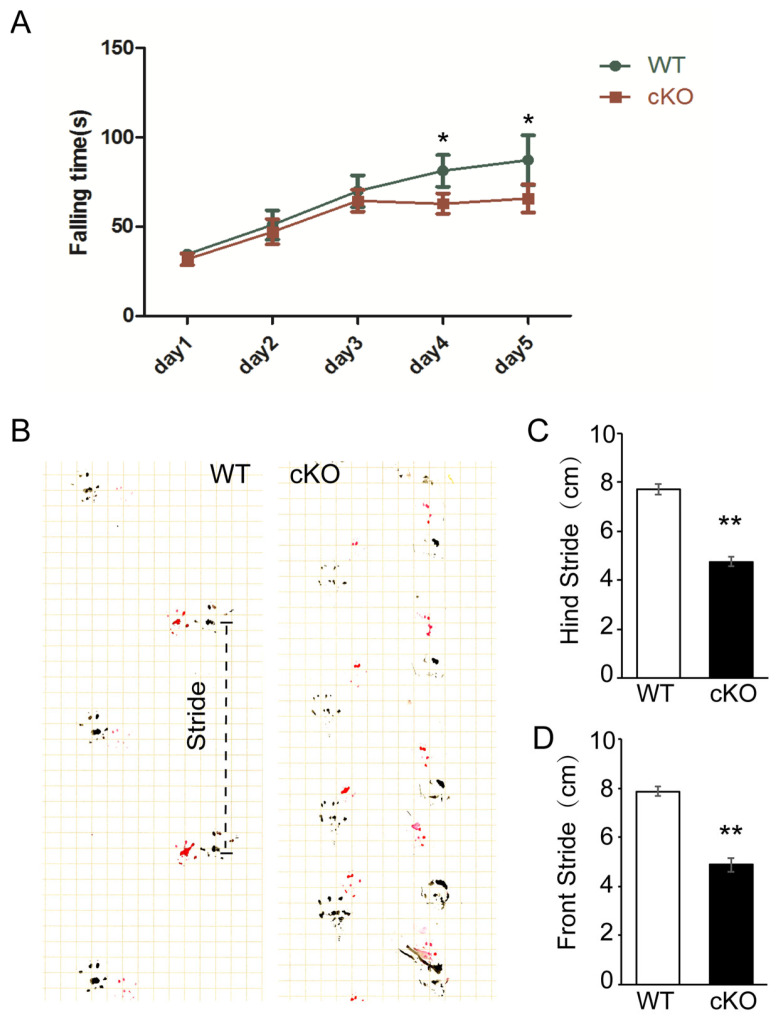
Thorase cKO mice exhibited impaired motor coordination. (**A**) The latency to falling of Thorase cKO mice and WT littermates at the age of 8 months (*n* = 7 males and 7 females). (**B**) Typical front (red) and hind (black) paw footprints of Thorase cKO and WT littermate mice at the age of 12 months (*n* = 3 males and *n* = 3 females). (**C**,**D**) Quantification of stride length as assessed by front stride length (**C**) and hind stride length (**D**). Red represents the forefoot and black represents the hind paw. Data are presented as the mean ± SEM determined using unpaired two-tailed Student’s *t* test. *, *p* < 0.05, **, *p* < 0.01.

**Figure 2 cells-12-02032-f002:**
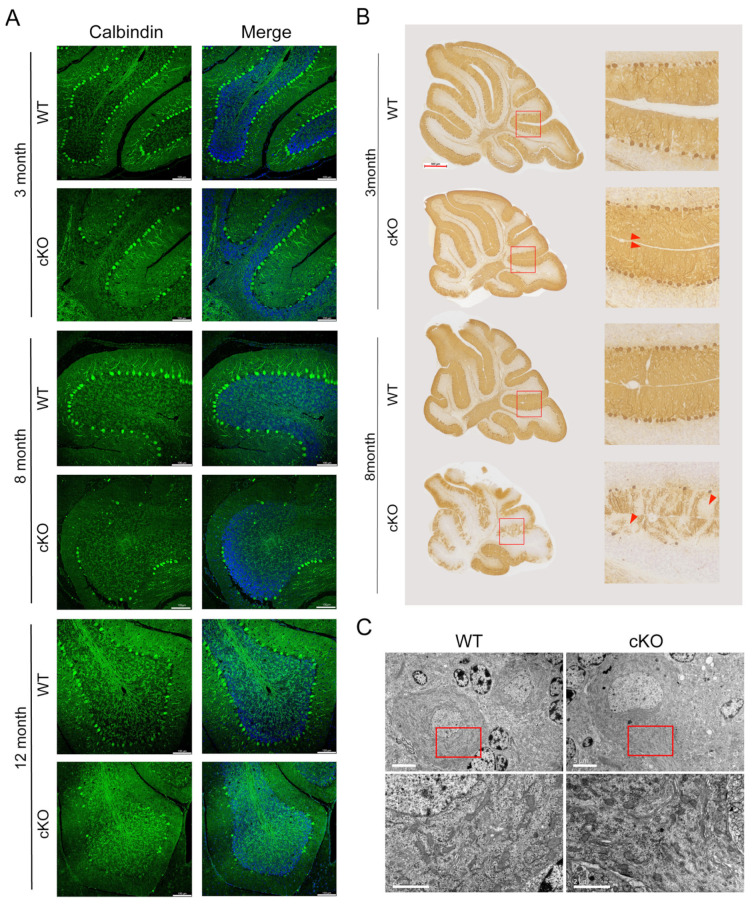
Conditional deletion in the cerebellum leads to degeneration and loss of Purkinje neurons. (**A**) Immunofluorescence staining with calbindin antibody for Purkinje cells in the cerebella of Thorase cKO mice and WT littermates at the ages of 3 months and 8 months. Scale bar, 100 μm. (**B**) Immunohistochemical staining with calbindin antibody to examine Purkinje neurons and dendritic fibers in Thorase cKO mice at 3 and 8 months. The red arrows indicate the dendritic fibers of Purkinje cells. Red rectangles represent for the corresponding right images of high magnification. Scale bar, 500 μm. (**C**) Typical morphological images of cerebellar Purkinje neurons and the mitochondria in 8-month-old Thorase cKO mice and WT littermate control mice observed using transmission electron microscopy. The figures below show the enlargement of the area. Red rectangles represent for the corresponding images below of high magnification. Scale bar, 5 μm (upper) and 2 μm (below).

**Figure 3 cells-12-02032-f003:**
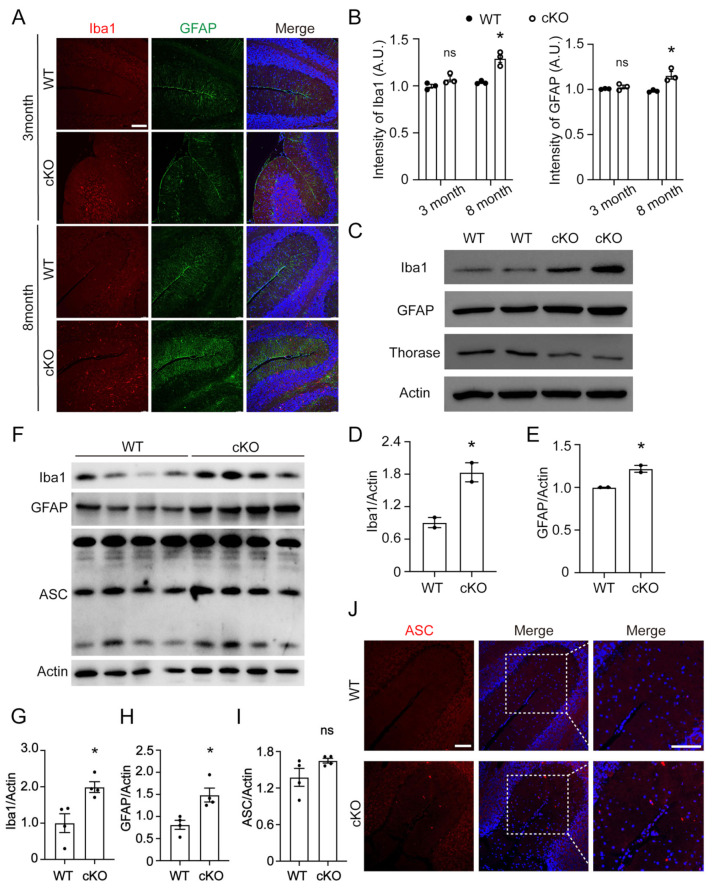
Thorase cKO mice exhibited an increase in activated microglia and astrocytes and inflammatory marker protein ASC expression. (**A**) Immunofluorescence staining with antibodies against Iba1 and GFAP to detect microglia and astrocytes in the cerebellar molecular layer of Thorase cKO mice and WT littermates. Scale bar, 100 μm. (**B**) Quantitative analysis of Immunofluorescence intensity of microglia and astrocytes in 3- and 8-month-old Thorase cKO and WT littermate brain tissue in panel (**A**). (**C**) Western blot analysis to detect the expression levels of the biomarkers Iba1 and GFAP in 8-month-old Thorase cKO and WT littermate mouse cerebellar tissue (*n* = 4). (**D**,**E**) Quantitative analysis of expression levels of microglia and astrocytes in 8-month-old Thorase cKO and WT littermate brain tissue in panel (**C**). (**F**) Western blots showing the expression levels of Iba1, GFAP and inflammation marker ASC in the cerebellum of 12-month-old Thorase cKO and WT littermates (*n* = 4). (**G**–**I**) Quantitative analysis of the expression levels of Iba1, GFAP and ASC in the cerebellum of 12-month-old Thorase cKO and WT littermates (**F**). (**J**) Immunofluorescence staining of ASC in the cerebellum of 8-month-old Thorase cKO mice and WT littermates. Scale bar, 100 μm. Data are presented as the mean ± SEM determined using unpaired two-tailed Student’s *t* test. n.s., not significance, *, *p* < 0.05.

**Figure 4 cells-12-02032-f004:**
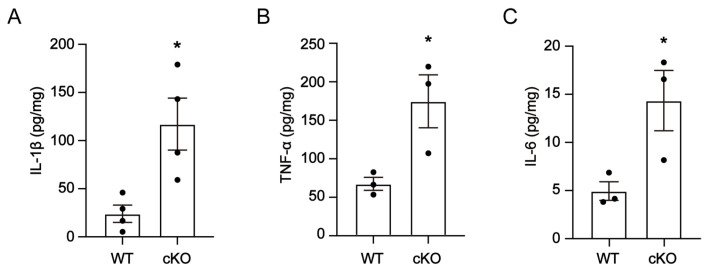
Thorase deficiency enhanced neuroinflammation in the cerebellum of Thorase cKO mice. (**A**) ELISA to detect the level of IL-1β in the cerebellum of Thorase cKO and WT littermates at the age of 12 months. (**B**) ELISA to detect the level of TNF-α in the cerebellum of Thorase cKO and WT littermates at the age of 12 months. (**C**) ELISA to detect the level of IL-6 in the cerebellum of Thorase cKO and WT littermates at the age of 12 months. Data are presented as the mean ± SEM determined using unpaired two-tailed Student’s *t* test. *, *p* < 0.05.

## Data Availability

Not applicable.

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
