# Peer review of "Genetic Deletion of Thorase Causes Purkinje Cell Loss and Impaired Motor Coordination Behavior"

_cells, 2023, doi:10.3390/cells12162032_

Round 1

Reviewer 1 Report

Dear authors,

After reading the manuscript I think it is of interest to researchers dealing with neurodegenerative diseases. The authors have clearly illustrated the results.

The literature consistent with the aim of the work was used and the conclusions demonstrated that Thorase, belonging to the AAA+ ATPase family, plays a fundamental role in maintaining cellular homeostasis. In fact, the data obtained on the in vivo model with Thorase silencing confirm that its absence leads to an increase in inflammatory proteins and neurodegeneration

The work can be published

Author Response

Dear Sir,

We thank you for your remarks as these are our central conclusions, what we consider our major discovery, and we thank you for your interest in the manuscript.

We would like to take this opportunity to appreciate all your efforts for the quality improvement of our manuscript.

Best regards,

Jianmin Zhang, Ph.D.

Professor, Department of Immunology

PUMC & CAMS

Reviewer 2 Report

The authors analyzed the role of Thorase in the cerebellum using a conditional knockout with Thorase specifically deleted from Purkinje cells. They showed the loss of Purkinje cells and the increase of astro- and microgliosis in the cerebellum. The paper is interesting and merits publication. However, I have several concerns that should be addressed in order to improve the quality of the paper.  

The authors omitted to incorporate the number of animals used for IHC, ICC, or Western

analyses.

2.5 Tissue Preparation

The authors state “The cKO mice and littermates WT mice of different months of age were injected…”. There is only 3 time-points analyzed in the paper, 3, 8, and 12 months-old animals. That should be noted.

20ug protein should be 20microg protein.

2.8 Immunofluorescence (IF) staining

Which sections were used for IF? What was the thickness of sections used?

Figure 3.

The staining for calbindin is not convincing. Images are blurry and the cell bodies are not stained at all. It seems like this antibody did not work properly. I suggest checking the staining. Considering that calbindin was used as a marker for Purkinje cells some insets and higher magnification images would be very useful.

Figure 4.

The IF staining for GFAP is not very convincing. The pictures appear blurry and the overall scarcity of astrocytes in the WT cerebellum is confusing. The 20% increase in the intensity of GFAP does not correspond to the presented images of GFAP staining.

The same concern stands for the IBA-1 staining.

I strongly suggest that authors check their antibodies and repeat the staining.

The IF staining for ASC depict ASC(+) cells in the cKO mice. The insets appear very blurry.  What are arrowheads pointing at in the wt in the Figure 4.F

What is the reason for doing western blot analysis at 8 and 12 months only for Iba-1 expression?

I am concerned that IF images, as presented, are not really improving the points that authors were trying to convey.

The manuscript should be proofread by the native  english speaking person.

Author Response

Please find the responses to the comments in the attached file.

Reviewer 3 Report

In principle the idea of this MS to make a conditional KO mouse to check for the function of a protein in a tissue or cell type is classical and intensively used. Thorase, the protein addressed in this study, is ubiquitously expressed in all mitochondria. There are several tissues and cells with increased energy demands, including Purkinje cells, which are known to be more affected by mitochondrial dysfunctions.

The PCP2 expressing Purkinje cells in which Thorase was conditionally knocked-down are affected only after more than 3 months in mice, with a clear cell loss after 8 months. While this phenotype could be interesting and exploited for understanding the age related neurodegeneration in mitochondrial-related diseases, this paper does not specifically address any of the related mechanisms and the presented data are very preliminary. 

The MS requires  extensive editing of English language. many sentences sound like being translated by Google translator, without any scientific and meaning checking.

Author Response

Please find the response letter to your comments in the attached file

Round 2

Reviewer 2 Report

The authors have adequately addressed the raised concerns.

Author Response

Dear Reviewer:

We appreciate your valuable comments for improving the quality of our manuscript.

Best regards,

Jianmin Zhang, Ph.D.

Professor, Department of Immunology

PUMC & CAMS

Reviewer 3 Report

The revised version of the MS has some improvements regarding the language and the description of the methods.

However, still many issues must be clarified/corrected.

Regarding the Title: “Genetic Deletion of Thorase Causes Purkinje Cell Loss and Motor Neurodegenerative Behavior”,  this paper shows no any degeneration of the motor neurons in the Thorase cKO mice, but of the Purkinje neurons. “Motor neurodegenerative behaviour” does not mean ataxic behaviour or impairment in motor coordination. The title should be in line with the main results.

The definition of ataxia is not clear:

“Ataxias is a kind of cerebellar neurodegenerative disease which manifests as clumsiness or imbalance of movement that is not caused by muscle weakness and is due to decreased function of the cerebellum, mainly resulting from degeneration of motor neurons or a variety of genetic mutations and traumatic injury [5–8]”

There are several problems in this definition. First, the term ataxia has the plural ataxias. There are many forms of ataxias that are not related to the degeneration of the motor neurons. Please find a better formulation from the cited literature.

Regrading the definition of the Purkinje cell layer:

“Purkinje neurons are a class of γ-aminobutyric acidergic neurons located in the middle of the granular and molecular layers of the cerebellum [8,9].”

The Purkinje neurons have their own layer, formed during development and organized as a monolayer in a complex process.

Regarding Fig 2 and the main findings contained in it:

In fig 2A, calbindin is not shown only in the dendrites of Purkinje cells, but also in the cell body.   There are not red arrow indicating to ”a typical seriously missing part of dendritic fibers”. The red arrows show the loss of the Purkinje cells only in (B) at 8 months.

Fig 2C does not show any clear “typical morphological images of Purkinje neurons” observed by transmission electron microscopy. The TEM pictures should be completed and the legend should be correct and clear. It is an important part of the paper to show degeneration at the organite level, but is it far to be well presented both in the legend and in the main text of the MS.

improved

Author Response

Dear Reviewer:

We appreciate your valuable comments for improving the quality of our manuscript. Please see the attachment.
